‘We like it wet’: a comparison between dissection techniques for the assessment of parity in Anopheles arabiensis and determination of sac stage in mosquitoes alive or dead on collection

Charlwood Jacques D. jdcharlwood@gmail.com 1
Tomás Erzelia V.E. 2
Andegiorgish Amanuel K. 3
Mihreteab Selam 4
LeClair Corey 5
1 Global Health and Tropical Medicine, Instituto de Higiene e Medicina Tropical, Universidade Nova de Lisboa , Lisbon , Portugal
2 MOZDAN , Morrumbene , Inhambane , Mozambique
3 Epidemiology and Public Health, College of Health Sciences , Asmara , Eritrea
4 National Malaria Control Program , Asamara , Eritrea
5 Medicines sans Frontiers , Bruxelles , Belgium
Negri Ilaria
Electronic publication date: 2018 Jul 10
Publication date: 2018
Volume: 6
Electronic Location ID: e5155
Received 2018 Mar 1; Accepted 2018 Jun 11
Copyright: ©2018 Charlwood et al.
Copyright year: 2018
Copyright holder: Charlwood et al.
License: This is an open access article distributed under the terms of the Creative Commons Attribution License, which permits unrestricted use, distribution, reproduction and adaptation in any medium and for any purpose provided that it is properly attributed. For attribution, the original author(s), title, publication source (PeerJ) and either DOI or URL of the article must be cited.
License URL: https://creativecommons.org/licenses/by/4.0/

Keywords: Anopheles arabiensis, Age-grading, Dissection, Virgin, Mated, Survival, Endophilic

Funding: Medical Research Council of Great Britain MR/L004437/1 Rowland The work in Tanzania was supported by the Medical Research Council of Great Britain (MR/L004437/1 Rowland). The authors received no funding for the work in Eritrea. The funders had no role in study design, data collection and analysis, decision to publish, or preparation of the manuscript.

==============================
Background

The determination of parous rates in mosquitoes, despite numerous shortcomings, remains a tool to evaluate the effectiveness of control programs and to determine vectorial capacity in malaria vectors. Two dissection techniques are used for this. For one, the tracheoles of dried ovaries are examined with a compound microscope and in the other the follicular stalk of ovaries is examined, wet, with a stereomicroscope. The second method also enables the sac stage of parous insects (which provides information on the duration of the oviposition cycle) and the mated status of insects to be determined. Despite widespread use the two techniques have not previously been compared.

Methods

We compared the two dissection techniques using Anopheles arabiensis, collected with a tent-trap in Eritrea. The paired ovaries were removed in water and one was examined by each method. From a separate set of dissections from Tanzania, we also determined if the sac stages of Anopheles gambiae s.l. (83% of 183 identified by PCR being Anopheles arabiensis the remainder being A. gambiae) that were alive on collection were different to those that died on collection and what the implications for vectorial capacity estimation might be.

Results

Seven per cent of the dry ovaries could not be classified due to granulation (yolk) in the ovariole that obscured the tracheoles. The sensitivity of the dry dissection was 88.51% (CI [79.88–94.35%]) and the specificity was 93.55% (CI [87.68–97.17%]) among the 211 ovaries that could be classified by the dry technique and compared to the ovaries dissected wet. 1,823 live and 1,416 dead from Furvela tent-traps, CDC light-trap and window-trap collections were dissected ‘wet’ from Tanzania. In these collections parous insects were more likely to die compared to nulliparous ones. The proportion of parous mosquitoes with ‘a’ sacs (indicative of recent oviposition) was significantly greater in insects that were dead (0.36) on collection in the morning compared to those that were alive (0.12) (Chi square 138.93, p < 0.001). There was a preponderance of newly emerged virgin insects in the outdoor collection (Chi sq = 8.84, p = 0.003).

Conclusions

In anophelines the examination of mosquito ovaries using transmitted light in a ‘wet’ dissection is a more useful and informative technique than examination of dry ovaries. In order to correctly estimate the duration of the oviposition cycle mosquitoes should be dissected as soon as possible after collection. Younger insects were more likely to attempt to feed outdoors rather than indoors.

Introduction

Despite its many shortcomings the measurement of parous and nulliparous rates (i.e., the proportion of insects in a population that have, or have not, laid eggs) in mosquito vectors is commonly evaluated as part of malaria control programs. The shortcomings include the requirement that measurements are made over a complete population cycle, that nulliparous and parous insects are sampled without bias and (for survival rate estimation) that survival is independent of age (Clements & Paterson, 1981). Although they may not provide much more than an approximation of survival (Gillies, 1989), an estimation of parous rates is useful in control trials for comparisons between intervention and control areas (where they are expected to be lower in interventions that target adult mosquitoes but higher in those that target larvae) and, independent of survival estimation, they also provide information on the behaviour of young insects that may themselves become a target for specific interventions.

Parity is determined by dissection. Following the maturation of the first batch of eggs irreversible changes occur in the ovaries of female mosquitoes. The tightly packed and coiled tracheolar system characteristic of nulliparous insects, illustrated in publications of the WHO (World Health Organization, 2013), becomes stretched and uncoiled as the eggs develop and never returns to its previous state (Detinova, 1962). In newly emerged teneral mosquitoes’ meconium, the remains of larval midgut epithelium can be seen as a green and opaque mass inside the midgut (Rosay, 1961; Romoser et al., 2000). This is excreted, either following an initial blood meal, or within 48 hrs of emergence. The tracheolar system can be seen in ovaries that are dissected in distilled water and allowed to dry. Once dry the ovary can be examined under a compound microscope. The ‘dry’ technique is simple and has been widely used.

An alternative technique is to examine the follicular stalk at the time of dissection with transmitted light using a stereomicroscope. The dissection is performed in isotonic saline (to avoid swelling of tissues) but can also be performed with water. This dissection has the advantage that it can provide information on both the mated status and the duration of the oviposition cycle in parous insects.

Irreversible changes also occur following oviposition in the pedicel that connects the ovarioles to the lateral oviduct (Hoc & Charlwood, 1990; Hoc & Wilkes, 1995). In this case granulation occurs in the basal body, small areas in the calyx wall enclosed by the ovariolar sheaths consisting of six to nine specialized epithelial cells, making them visible when examined with light coming through the preparation. A large egg sac remains in the ovarioles immediately after ovulation. The sac gradually contracts and, 12–24 h after ovulation, consists of heavily folded tunica above the calyx. In nulliparous females, there is no coloration of the pedicel. The tracheolar system is visible in this dissection so that its appearance can also assist in interpretation of the preparation. Examination of the pedicel in parous females also enables the duration of the oviposition cycle to be estimated since an insect with a large sac would have been caught shortly after oviposition whereas one with just a basal body would have oviposited approximately a day earlier. The duration of the oviposition cycle has a major impact on the proportion of mosquitoes that might be vectors. For example, a change from a two-day cycle to a three-day one produces a four-fold increase in the potential numbers of vectors.

The ‘wet’ dissection also enables the mated status of newly emerged nulliparous insects to be determined. In particular, examination of the spermatheca and oviducts is possible. Virgin insects do not have sperm in the spermatheca and, in recently mated anophelines, there is a male donated mating plug (Gillies, 1956; Baldini et al., 2013) visible in the common oviduct. This is absorbed over the following 12–24 h. Thus, with this dissection it is possible to separate nulliparous insects into three categories: virgins, recently mated insects and those that have mated 24 h, or more, earlier (Charlwood et al., 1985; Charlwood, Birley & Graves, 1986; Charlwood et al., 1986; Charlwood et al., 1997; Charlwood et al., 2003a; Charlwood et al., 2003b; Charlwood et al., 2011; Charlwood & Tomás, 2011). In practise, the overall appearance of the ovary is used to assess parous status: transparent and small ovaries with coiled tracheoles are indicative of nulliparity, whilst larger, darker, ovaries with enlarged ampullae (Gillies, 1956) and an uncoiled tracheolar system indicate that the mosquito is parous.

Anophelines and Aedines differ in that in the former, oogenesis is an ‘all or nothing’ phenomenon that requires a complete blood meal to proceed, whilst in the latter, individual follicles may develop following partial blood meals. This makes estimates of age more difficult in the latter group.

Hugo et al. (2008) compared these and more sophisticated techniques using laboratory reared Aedes vigilax and Culex annulirostris. They considered that the dry technique (when allied to the observation of the presence of meconium in the stomach of the mosquito) was the most suitable for parity determination.

The two methods of dissection have not previously been compared in anophelines, nor have they been compared using wild insects whose life conditions differ from insects reared and maintained in the laboratory. Here, therefore, we compare these methods with wild caught Anopheles arabiensis. We also compare estimates of the duration of the oviposition cycle from insects that died shortly after collection with those that remained alive up to the time of dissection.

Methods

Description of study sites

Anopheles arabiensis, collected on 17 nights between the 7th and 23rd of October 2017 with a Furvela tent-trap (Charlwood et al., 2017) below the village of Adi Bosco (15°41′41.67″N 38°38′54.59″E at an altitude of 1,536 m above sea level) in Anseba province, Eritrea, were dissected in bottled drinking water. Mosquitoes, that were alive upon collection, were killed in a freezer and were then used for these dissections. One ovary was placed on a slide to dry for subsequent examination and the other was assessed directly for parity and sac stage. Insects from the latter dissection were classified according to the scheme outlined by Charlwood et al. (2003a) and Charlwood et al. (2003b). The sac stage in parous insects was determined according to the scheme outlined in Wilkes & Charlwood (1979) and illustrated in Fig. 1.

Figure 1 Sac stages of parous mosquitoes.

Sac stages (a, b, c, d) as seen in host-seeking dissected mosquitoes. ‘a’ and ‘b’ sacs refer to mosquitoes that are considered to have oviposited a few hours before collection/dissection whilst ‘c’ and ‘d’ sacs are found in mosquitoes that have delayed returning to feed. g, germarium; o, ovariole; s, sac; lo, lateral oviduct.

Each mosquito was given a unique identifying number and subsequent comparisons between assessments of parity were determined. A number of Culex quinquefascaitus collected with a CDC light-trap from a bedroom in Asmara, Eritrea, where potential hosts slept under mosquito nets, were also dissected for a comparison of the appearance of the ovaries of the two species.

The sac stages of mosquitoes according to whether they were alive or dead upon collection were determined from 42 nights of collection undertaken in the village of Kyamyorwa in Muleba district, Kagera Province, Tanzania, from December 1 2015 to January 17 2016. Mosquitoes were collected in a CDC light-trap, run inside a bedroom with two human and one canine host; a window-trap from the same room and a Furvela tent-trap outdoors with a single sleeper (LeClair et al., 2017). Live mosquitoes were removed from the collection bags with an aspirator prior to being killed and both recently killed and those dead on collection were identified to species or species group using the keys of Gillies & De Mellion (1968) and Gillies & Coetzee (1987). Mosquitoes in Tanzania were dissected in saline eye drops (Charlwood et al., 2017).

Anopheles arabiensis is the only member of the A. gambiae complex that has been identified from previous collections in Eritrea (Shililu et al., 2003) and so it is assumed that this was the member of the complex that was collected. A sub-sample of the A. gambiae s.l. from Kyamyorwa were identified to species by multiplex real-time PCR Taq Man assay (Bass, Williamson & Field, 2008).

In order to determine if the different age groups were caught in similar proportions indoors (in light-trap and window-trap combined) and outdoors (in the tent-trap) the number of the different ages collected live and dead were estimated by multiplying the total by the proportion in each category and then summing the estimated totals. The overall proportion of each age group (indoors and outdoors) was then estimated and indoor and outdoor collections compared by Chi-Square test (at a significance level of 0.05).

We also assume that gonotrophic development (from blood feeding to becoming gravid) takes two days in Kyamyorwa; hence mosquitoes with ‘a’ or ‘b’ sacs (Sac) were considered to have a two-day feeding cycle and those with ‘c’ or ‘d’ (No-sac) to have added an extra day (i.e., to have a three-day cycle). Estimates of the population mean duration of the feeding cycle (µ) in live and dead parous insects were therefore determined according to the proportions of Sac and No-sac mosquitoes in the collection where µis the mean feeding frequency of parous insects in days: μ=nSac∗2+nNo-sac∗3∕nSac+nNo-sac.

The sensitivity (true positivity) and specificity (true negativity) of the dry dissection was determined using the software MedCalc (https://www.medcalc.org/calc/diagnostic_test.phpto). For the purposes of comparison the wet dissection was considered to be the ‘gold’ standard.

Ethics

The collections conducted in Tanzania were done as a component of the Pan African Malaria Vector Research Consortium project ‘Evaluation of a novel long lasting insecticidal net and indoor residual spray product, separately and together, against malaria transmitted by pyrethroid resistant mosquitoes’ which received ethical clearance from the ethics review committees of the Kilimanjaro Christian Medical College (certificate number 781 on 16 September 2014), the Tanzanian National Institute for Medical Research (20 August 2014), and the London School of Hygiene and Tropical Medicine (reference 6551 on 24 July 2014). The trial was registered with ClinicalTrials.gov (registration number NCT02288637) on 11 July 2014.

Collections in Eritrea were undertaken by the first author in his tent during supervision of students from the College of Health Sciences, Asmara, undertaking their fieldwork as part of a course entitled ‘The ecology of malaria vectors’.

Results

Ovaries of Cx. quinquefasciatus were clearer and the tracheoles easier to see than was the case with the A. arabiensis (compare Figs. 2A, 2B with Figs. 3A, 3B).

Figure 2 Photograph of dry ovaries of Culex quinquefasciatus.

Dry ovaries of Culex quinquefasciatus showing their archetypal, ‘textbook’ appearance (A) nulliparous female showing coiled tracheoles; (B) parous female showing the tracheoles now streched and uncoiled.

Figure 3 Dry ovaries of Anopheles arabiensis (A) nulliparous; (B) parous.

Compare these with the Cx. quinquefasciatus ovaries in Fig. 2.

In almost 10% (23 of 238) of the A. arabiensis dissected dry, the deposition of yolk in the follicles made assessment of the age difficult or impossible from the dry dissections. In some cases, wetting the preparation again temporarily enabled the tracheoles to become visible for assessment (Figs. 4A and 4B).

Figure 4 Ovary of mosquito with yolk (A) before and (B) after adding water to the dry preparation.

Some dry ovaries that contain yolk (making interpretation difficult) can be temporarily improved by the addition of water to the preparation. The figure shows one such ovary. (A) dry; (B) with water added.

Comparison between methods

There were 211 ovaries that could be classified by the dry technique and compared to the ovaries dissected wet (Table 1). There was a 91.5% (CI [86.30–94.49%]) concordance between the methods. Nevertheless, 18 of 211 (10 parous and 8 nulliparous) were given different classifications by the two methods. Thus, assuming that the wet dissection was the ‘gold standard’, the sensitivity of the dry dissection was 88.51 % (CI [79.88–94.35%]) and the specificity was 93.55% (CI [87.68–97.17%]).

Table 1 Age of Anopheles arabiensis determined either by immediate ‘wet’ dissection using transmitted light or examined dry with a compound microscope.

		Dry dissection	
		Nulliparous	Parous	Total	
Wet dissection	Nulliparous	116	10	126	
Parous	8	77	85	
Total	124	87	211	

During the experiment, the number of A. arabiensis collected decreased from a mean of 126 per tent per night to 34 per night whilst the parous rate increased from 0.28 to 0.56 (correlation between the number collected and parous rate = −0.71). Since the population was changing and collections did not cover the complete population cycle any estimates of survival from the present data would be imprecise and possibly incorrect.

Sac stages among live or dead mosquitoes

Among 183 A. gambiae s.l. from Kyamyorwa identified to species by PCR 152 (83.1%) were A. arabiensis and the remainder were A. gambiae (LeClair et al., 2017). Thus, the great majority of insects from Kyamyorwa were also A. arabiensis.

Between November 30 2015 and January 17 2016, 1,823 live (273 from the light- trap, 341 from the window trap and 1,209 from the tent-trap) and 1,416 dead (711 from the light-trap and 705 from the tent-trap) A. gambiae s.l. were dissected (Table 2). The smaller numbers of live insects dissected from the light-trap was due to the low survival of the mosquitoes in the trap (LeClair et al., 2017). All insects collected from the window trap were alive. 574 (29.9%) of the live insects and 689 (39.9%) of the dead insects were parous (Chi-Square 10.03 p = 0.002). Thus, parous insects were more likely to die compared to nulliparous ones. Among the 1,281 live nulliparous insects dissected, 567 (44.26%) were virgins whilst 457 (35.68%) had mating plugs (Chi-Square 5.44, p = 0.020). Among the dead nulliparous insects dissected 311 (36.76%) were virgins and 366 (43.26%) had mating plugs. The estimated total proportion of the different age groups (combining estimated numbers of both live and dead insects) was also different between mosquitoes collected indoors (light and window-trap combined) and those collected outdoors (Table 3). Virgin insects predominated in the outdoor collection (Chi-Square 16.54. p < 0.001) whilst parous insects, even excluding teneral insects (virgins and those with mating plugs), were more common indoors (Chi-Square for all insects 41.96, p < 0.001 and 8.84, p = 0.003 excluding teneral insects). Hence, newly emerged insects were more likely to attempt to feed outdoors rather than indoors.

Table 2 Number of A. arabiensis dissected by age, collection type and mosquito condition (live or dead) on collection.

		Virgin	Plug	NI	NII	a-sac	b-sac	c-sac	d-sac	Total dissected	Parous rate (Adj Wald C.I.)	
Tent	Live	385	321	56	127	32	51	77	160	1,209	0.27 (0.25–0.29)	
	Dead	180	167	31	57	94	64	56	56	705	0.38 (0.35–0.42)	
Light	Live	107	58	6	29	10	7	13	43	273	0.27 (0.22–0.32)	
	Dead	131	199	31	50	114	82	64	40	711	0.42 (0.39–0.46)	
Window	Live	75	78	10	29	24	34	36	55	341	0.44 (0.39–049)	
Notes.

Note all mosquitoes in the window trap were alive on collection.

TITLE Virgin mosquitoes without sperm in the spermatheca

plug mosquitoes with a mating plug in the common oviduct

NI nulliparous mosquitoes with sperm in the spermatheca but no mating plug visible and with ovarioles without yolk

NII nulliparous mosquitoes with sperm in the spermatheca, without a mating plug visible and with ovarioles with yolk present.

Adj. Wald C.I. Adjusted Wald 95% Confidence Interval

Table 3 Number of A. arabiensis collected indoors (light-trap and window-trap) and outdoors alive or dead on collection and proportion in each age category.

Location	Condition	Total collected	Propn Virgin (Adj Wald C.I.)	Propn Plug	Propn Null	Propn Parous	
Indoora	Live	560	0.27 (0.23–0.31)	0.22 (0.19–0.26)	0.62 (0.58–0.66)	0.38 (0.34–0.42)	
Dead	3,865	0.22 (0.19–0.25)	0.25 (0.23–0.29)	0.58 (0.54–0.61)	0.42 (0.39–0.46)	
All	4,425	0.22 (0.21–0.23)	0.25 (0.24–0.26)	0.58 (0.57–0.60)	0.42 (0.40–0.43)	
Outdoor	Live	2,029	0.30 (0.28–0.33)	0.27 (0.25–0.29)	0.73 (0.71–0.75)	0.27 (0.25–0.29)	
Dead	1,605	0.24 (0.21–0.27)	0.23 (0.21–0.26)	0.62 (0.59–0.65)	0.38 (0.35–0.41)	
All	3,634	0.27 (0.26–0.29)	0.25 (0.24–0.27)	0.68 (0.67–0.70)	0.32 (0.30–0.33)	
Notes.

a Indoor includes light-trap and window-trap collections combined.

Kyamyorwa, Tanzania, December 2015–January 2016.

TITLE Propn proportion

Adj. Wald C.I. Adjusted Wald 95% Confidence Interval

Virgin insects without sperm in the spermatheca

Plug insects with a mating plug in the common oviduct

Null All nulliparous insects combined; Parous - all parous insects combined

Among parous insects the proportion with ‘a’ sacs was significantly greater in insects that were dead (0.36) on collection in the morning compared to those that were alive (0.12) (Chi-Square 138.93, p < 0.001) (Fig. 5). The estimated duration of the oviposition cycle among live insects, based on equation 1, was 2.7 days and among dead ones was 2.4 days. The proportion of parous insects dissected from Adi Bosco (that were all alive on collection) with large sacs was also significantly different to those from Kyamyorwa (68 of 91 compared to 113 of 424) (Chi-Square 75.97, p < 0.001).

Figure 5 Sac stages among Anopheles arabiensis that were either alive or dead upon collection.

There is a greater proportion of Anopheles arabiensis females with ‘a’ or ‘b’ sacs among insects that had died upon collection whilst among those that had remained alive the sacs had contracted to ‘c’ or ‘d’ by the time they were collected.

Parous rates were lower in the mosquitoes that had remained alive at the time of capture (Chi-Square = 39.46, p < 0.05). There was no significant difference in the parous rates of mosquitoes collected in the window trap compared to the light-trap (Chi-Square = 2.57, p = 0.109 n.s.) nor between virgin and plug rates among newly emerged insects from these two types of collection (Chi-Square = 0.0002, p = 0.98. n.s.) but there was a difference between tent and window trap (Chi-Square = 21.76, p =  < 0.001).

Discussion

Reviews of techniques for age determination in insects of medical importance, including the methods used in the present study, have been presented by Charlwood, Rafael & Wilkes (1980) and Tyndale-Biscoe (1984). Ovaries of Cx. quinquefasciatus were much easier to classify using the dry technique than were the A. arabiensis. The dry technique may therefore be useful for determining the gonotrophic age of this mosquito. Our results indicate, however, that almost 10 per cent of the A. arabiensis had unreadable ovaries using this technique, which might affect assessments of survival. Nevertheless, since the discrepancies were almost equally distributed between nulliparous and parous insects an overall estimate of survival would be similar. A similar proportion of unreadable ovaries of Aedes vigilax and Culex annulirostris was observed by Hugo et al. (2008). As with the A. arabiensis this was apparently due to the deposition of material (yolk) in the follicles that obscured the tracheoles. This may particularly be a problem with pre-gravid insects that have taken a blood-meal and in which the follicles have advanced to mid to late Stage II. Thus, despite its ease, the dissection of Anopheles, ovaries in water and their subsequent examination with a compound microscope when dry, is not as good, or useful, as examination of the ovaries using a stereomicroscope with transmitted light from a mirror. A mirror is better than an artificial light source since by altering its position, the contrast of the preparation can be changed so that the visibility of structures within the ovaries changes making assessment easier.

Results from Tanzania indicate that young A. arabiensis, particularly virgin insects, are more likely to feed outdoors than older ones. This is similar to the behaviour of Anopheles coluzzii from Ghana (Charlwood et al., 2012) and indicates that mating has an effect on host seeking in a relatively subtle fashion, at least in the A. gambiae complex but, whether the same behaviour occurs in other endophilic mosquitoes is not known.

In many ways young virgin insects act like surrogate males. Thus, they may be more likely to rest in the sites that males do (i.e., outdoors in the case of the A. gambiae complex) compared to older females. Many mosquitoes rest close to where they feed. Host-seeking mosquitoes respond to visual as well as olfactory cues (Bidlingmayer, 1975; Bidlingmayer & Hem, 1979; Hawkes et al., 2017). Young insects may have different visual priorities (horizontal swarm markers) and may not respond to the silhouette of the house in the same way that older mosquitoes do. This would lead to there being a preponderance of young insects biting outdoors. In the absence of control measures (present throughout most of the insects evolution) there is little mortality among mosquitoes that feed and rest indoors. For virgin insects to spend their time outdoors there must be a survival advantage. It may mean that such insects are more likely to find a mate compared to insects that rest indoors.

Whilst a preponderance of young insects biting outdoors might mean that the risk of acquiring malaria per bite is lower outside it also means that the risk of transmitting are greater outdoors since young mosquitoes may be more likely to survive through the extrinsic cycle than older ones. They also imply that a potential control technique aimed specifically at young insects should work preferentially outdoors. Young, naïve mosquitoes may be attracted to a wider range of potential hosts than older insects, which may return to feed on hosts that they have fed on previously (Vantaux et al., 2014; Vinauger, Lutz & Riffell, 2014). Odour baited traps that target such young insects may be one possible approach.

The proportion of live parous mosquitoes with ‘a’ sacs from the tent-trap recorded from Eritrea was significantly higher than that recorded from Kyamyorwa. The higher rates are probably because the much lower temperatures in Adi Bosco (12 °C minimum at night in Adi Bosco compared to 27 °C in Kyamyorwa) slowed contraction of the sacs. At the higher temperatures, typical of the tropics, it behoves the entomologists to kill and dissect the mosquitoes as soon as possible after collection. If there is a delay, sacs are likely to contract during the time that the mosquito is collected and killed. This will tend towards an overestimation of the duration of the cycle (in our case 2.7 days compared to 2.4 days) and as a result an overestimation of the vectorial capacity of the population as a whole. Given the variation in age and the effect that environmental conditions can have on the relative proportion of the population biting indoors or outdoors (Charlwood & Tomás, 2011) it also behoves the entomologist to undertake simultaneous collections indoors and outdoors for population assessment.

Surprisingly, virgins survived better than recently mated insects. This may be because they were collected later in the night than recently mated insects (and so had a shorter time in the stressful environment of the trap). However, given that virgin and recently mated females of A. coluzzii have similar patterns of activity in landing collections (Charlwood et al., 2003a; Charlwood et al., 2003b) and that the rates were similar between light-trap (where the majority of mosquitoes had died) and window-trap (where they were all alive, LeClair et al., 2017) this is unlikely. Given that it had not been absorbed before dissection, it is unlikely that possible nutritional benefit to the female from the male donated mating plug was available. Under these circumstances the effort involved in mating may have had a detrimental effect on the survival of the female.

It is possible that dissections will in the future be replaced by other techniques, notably assessment of age based on reflectance of Near Infra-Red (NIR) light (Mayagaya et al., 2009; Krajacich et al., 2017) or gene transcription (Cook et al., 2006; Cook et al., 2007). Nevertheless, the techniques remain experimental and in the process of development. For the time being dissections remains the method of choice.

Conclusions

The utility of examination of tracheolar coiling in dried ovariolar dissections for the assessment of mosquito age differs between genera. Among anophelines the technique is less useful than examination of ovaries wet with transmitted light.

The wet dissection also allows for determination of oviposition cycle duration. However, insects need to be dissected shortly after capture for the information to be meaningful.

Recently emerged virgin Anopheles arabiensis are more likely to be seek hosts outdoors rather than indoors.

Supplemental Information

Data S1 Raw data

Click here for additional data file.

We would like to thank the staff of the entomology laboratory of Elaboret for their accommodation and help during the studies in Eritrea and Yohannes Kulwa and the late Mzee Kasege and his family for their help during the work in Tanzania. We also thank Enock Kessey for the identification of the A. gambiae complex mosquitoes from Tanzania. We would like to thank the referees and Editor for their perceptive comments that helped improve the manuscript. Thanks to Sarah Moore and Mark Rowland for their comments regarding the behavior of virgin females.

Additional Information and Declarations

Competing Interests

Author Contributions

Data Availability

The authors declare there are no competing interests.

Jacques D. Charlwood conceived and designed the experiments, performed the experiments, analyzed the data, contributed reagents/materials/analysis tools, prepared figures and/or tables, authored or reviewed drafts of the paper, approved the final draft.

Erzelia V.E. Tomás conceived and designed the experiments, performed the experiments, approved the final draft.

Amanuel K. Andegiorgish contributed reagents/materials/analysis tools, approved the final draft, provided logistical backup in the field.

Selam Mihreteab approved the final draft, provided logistical backup in the field.

Corey LeClair conceived and designed the experiments, analyzed the data, authored or reviewed drafts of the paper, approved the final draft.

The following information was supplied regarding data availability:

The raw data are provided in Data S1.

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
