# Peer review of "‘We like it wet’: a comparison between dissection techniques for the assessment of parity in Anopheles arabiensis and determination of sac stage in mosquitoes alive or dead on collection"

_PeerJ, doi:10.7717/peerj.5155_

## Round 0.1 · original submission · Major Revisions

This manuscript received three reviews. While two reviewers were positive or made requests for only minor revisions, the first raises significant concerns mainly involving the methods used by the authors with the suggestion to repeat the whole study in more controlled conditions.

I personally think that there is no need to replicate the study because it actually provides some evidences that one method (wet dissection) may be more useful and informative that the other (dry dissection). However, I also think that comments of reviewer 1 should be considered, for example those regarding the possibility that field conditions might affect the status of the mosquitoes’ ovary or that a spermatheca full of sperms gives an "actual" probability that the female will ovideposit.

Here are some other specific comments:

- results:
please check concordance in number of dissected specimens
- discussion:
please provide discussion on some results (e.g. increased number of potential vectors following changes from 2-day to 3-day cycles);
please consider also possible sampling bias in result interpretation, as well as the dissection ability;
please discuss why virgins may survive better, why newly emerged mosquitoes are more likely to attempt to feed outdoors rather than indoors and if recently oviposited females are more likely to die than others;
please discuss also the possible impact of the pregravid blood meal on using the dry method.
Finally, authors should also include in the discussion ecological data on the mosquito species which can help data interpretation.

Reviewer 1 ·

Basic reporting

see below

Experimental design

see below

Validity of the findings

see below

Additional comments

The manuscript of Charlwood and colleagues present a comparative evaluation of two microscopy techniques for the determination of parity status of Anopheline mosquitoes and other basic information about their population dynamics. The English is appropriate but needs to be revised, as I noticed several typos in the text. The manuscript, despite its main question is worth to be investigated, shows several limits and needs to be improved in many aspects. First of all, I don’t agree with the approach to use only field specimens to assess the quality of the two techniques: the field conditions, which are by definition unknown, are not ideal to evaluate the best method to assess the parity of the mosquitoes, as some unknown environmental/physiological factors could have affected the status of the ovaries. Moreover, the test is based on the assumption that the wet dissection is the gold standard but no justification are provided for this choice. The comparison of two techniques should be based on the postulation that one of the two is the best known approach in order to assess the outcome to be detected. The authors should provide bibliographic information to support their choice. Anyway, there are no sure information about the oviposition event of the field mosquitoes collected. The only added information could be the presence of sperm in the spermathaeca, which could give information about mating but no guarantees that an oviposition event happened or not. So, how the authors are sure that the mosquitoes were parous or nulliparous? An insectary study based on mosquitoes that have done single oviposition as positive control would have been a better approach. With the protocol adopted in this study it is impossible to determine the false positives and false negatives, which is the basis of the determination of sensitivity and specificity of a technique compared to another. For all these reasons, I strongly suggest to repeat the study also in insectary conditions in order to have a complete picture.
Other comments:
- in methods section it is not reported how concordance, specificity and sensitivity is calculated, please describe the formulas citing the appropriate references.
- lines 192-193 (and also 250-251): it is not clear to me what is the aim of the comparison between Culex quinquefasciatus and Anopheles arabiensis performance of the techniques. What is adding this test to the manuscript? Moreover, the better quality of the former species is not based on objective evaluation and is not described how a reader could evaluate this quality in order to use this result for other studies. I suggest to better integrate this experiment or to remove it from the manuscript.
- line 194: the numbers of A. arabiensis dissected here is 238, while in the abstract is reported 389 collected. So, why it is not included the number of mosquitoes collected in Results? Moreover, what about the remaining 151 mosquitoes?
- lines 200-201: the 286 mosquitoes collected are from which site? In table 1 is reported a total of 289 mosquitoes, while here 211 are reported. Please correct.
- lines 206-212: This is a comment that should be moved into Discussion. Also, it could be interesting to specify how many nights of collection have been done, as the total number of mosquitoes should be 389. I suppose that the number of nights would be not more than 3-4. So, it is possible that the difference in parous rate is just due to a sampling bias. The authors should take into account this possibility in the interpretation of this result.
- lines 215-217: please describe the other species of the A. gambiae complex found (I suppose just A. gambiae s.s.)
- lines 218-220: the numbers in the text and in table do not seem to match (3649 in text, 3239 in table 2). Please check.
- line 225, 242, 244, 246, 247: please add the percentage of mosquitoes used for the Chi-square test.
- Discussion: I don’t find any specific comment on the conclusion of the authors about the results of wet vs. dry technique. Which one is the best?
- lines 262-265: the proportion of out vs in collected virgin females could be just a consequence of the interception of newly emerged mosquitoes and not a behavoiural effect due to mating.
- figures 2-3-4: please add the captions
- table 1: please add the caption
- table 2: please describe better the table: add “collection method” to the first column and “status” to the second one. Please describe the acronyms in caption.
- tables 2 and 3: which method has been eventually chosen to define the parous rate?

Reviewer 2 ·

Basic reporting

The manuscript is clearly written. The background was addressed appropriately, methods were described concisely and results presented in an easily understandable manner.
There is one section which is difficult to follow and can be improved (see other comments to the author).

Experimental design

The experiment was well designed and methods explained with sufficient detail to allow its replication. The comparison of the two methods presented is of importance to researchers and health authorities wanting to measure mosquito survival in the field and therewith evaluate the impact of vector control interventions. The rationale was well described in the introduction. Although dry dissections are considered "easier" the wet method has been thought to have higher accuracy. The results presented in this article indicate dry and wet dissections provide similar information in regard to parity status of a malaria mosquito. Ethics approval was sought.

Validity of the findings

Conclusions are well stated.

Additional comments

Line 101: “For example, a change from a two-day cycle to a three-day one produces a four-fold
increase in the potential numbers of vectors.” This is very interesting, could the authors please clarify how this conclusion was reached? Increasing from two-day cycle to three-day cycle would also reduce the number of potentially infected blood meals a mosquito would take along its life.

Line 167 The authors talk about different types of sacs (a,b,c) it is not clear what these are. Please define them earlier in the text.

Line 166 – 171 This section is difficult to follow. It seems like there is some important information missing. Please explain what the authors mean by the “sac” “no sac” mosquitoes, is this the egg sac? Isn’t the presence of a sac just indicative that eggs have been laid recently? How can the length of the gonotrophic cycle be inferred from this observation?

Line 204 One of the clear limitations of dissections is the level of dexterousness needed to perform the technique and the difficulty of standardizing the technique since it depends on each individual’s talent. Also, there is an arguable degree of subjectivity. This should be mentioned given that sensitivity and specificity of the technique will vary according to whoever is performing the dissections.

Line 216 “Among 183 A. gambiae s.l. from Kyamyorwa identified to species by PCR 152 (83.1%) were A. arabiensis” What were the remaining 17%?

Line 224 “Among the nulliparous insects, virgins survived better than those with mating plugs (Chi-Square 5.4373, p = 0.020).” Do the authors think there is a biological explanation to this finding?

Line 231 “Hence, newly emerged insects were more likely to attempt to feed outdoors rather than indoors.” I believe this finding has been seen in other studies however there is no consensus why younger mosquitoes prefer to feed outdoors, do the authors have a theory? Is it relatable to “experience” of the vector? Do the authors think anophelines can learn and remember? I think this is a very interesting topic that isn’t discussed enough. Perhaps the authors would like to share their hypothesis backed up by their data.

Line 234 “Among parous insects the proportion with ‘a’ sacs was significantly greater in insects that were dead” So is this indicative that females that have recently oviposited are more fragile and likely to die than others?

Line 298 I would say the gold standard rather than the method of choice.

Reviewer 3 ·

Basic reporting

Well prepared paper, clear and concise.

Literature cited left the author to conclude that they were the first group to use both dried and wet preparations to determine age in anophelines. While this paper may be the first to compare each method as a stand alone assessment of parity, several previous papers have combined both methods to ascertain parity [dry methods] and number of ovipositions [wet method].

Experimental design

Well presented.

The only flaw in the work, is that they really don't know which method is the gold standard. The senior author [a well respected expert] selected the wet method based on his experience. It may have been useful to prepare a known set of standards similar to Hugo et al.

Validity of the findings

It might have been interesting to present the data as 2-way tables comparing nulliparous and parous by each method to show where the difference lie. This was addressed statistically on L204-205.

Additional comments

As discussed by the authors, estimates of parity and therefore survivorship can be biased by the collection methods used, time specimens were held and source of the material dissected. The authors did not discuss the impact of the pregravid blood meal on using the dry method. Here, the small blood meal often taken by virgin females in lieu of sugar advances the ovarioles to Stage IIa and this yolk can make it difficult to see the tracheoles. It also is important to understand the ecology of the species. Some anopheles rest indoors but egress to feed outdoors; they also oviposit and refeed the same night, and rapidly transition from sacculate to dilation. If they fail to find a blood meal the same night, they will appear parous but not sacculate the following evening.

---

## Round 0.2 · accepted · Accept

I think that the manuscript is now ready to be published on PeerJ.

# Reviewer 1 ·

Basic reporting

see below

Experimental design

see below

Validity of the findings

see below

Additional comments

All comments have been addressed. The manuscript is acceptable as it is.